# Determining the perceived acceptability of an intervention designed to improve health literacy around developmentally appropriate play during infancy, with a community advisory group of mothers, in Soweto, South Africa

Fiona Bennin[ID]*, Helene Theunissen, Shane A. Norris, Alessandra Prioreschi

SA MRC/Wits Developmental Pathways for Health Research Unit, Department of Paediatrics, Faculty of Health Sciences, School of Clinical Medicine, University of the Witwatersrand, Johannesburg, South Africa

* fionabennin@gmail.com

**Data Availability Statement:** All data can be found in the manuscript and Supporting information files.

## Abstract

Children require opportunities to participate in unstructured, unrestricted active play as infants, to encourage positive health, socioemotional and developmental outcomes in life. Certain social and environmental factors in the home setting can limit participation in play, particularly in low resource settings. As caregivers are their infants' first teachers, they have the important role of providing opportunities and space for children to learn through play. This mixed methods, cross-sectional study aimed to test the acceptability of an intervention developed to improve health literacy around play and development with mothers from Soweto, South Africa. Fifteen mothers with infants aged 0–3 months were included to form a Community Advisory Group (CAG). Two rounds of focus group discussions (FGDs) were conducted to explore the understanding of, and barriers to play and development, and to determine the acceptability of an intervention prototype. The prototype included developmentally appropriate activities presented in video format, demonstrations of how to make toys, infographics and other educational material which would be delivered to participants as part of the intervention. A further questionnaire was administered to participants one week after the FGD to determine the acceptability of intervention activities which the participants completed at home, with their infants. Participants reported several barriers to play, including limited options for safe outdoor play, overcrowding, insufficient time, limited resources, and conflicting information. Participants suggested that the intervention content be delivered every 1–2 weeks, through a data-free app. Overall, the prototype activities tested at home were deemed acceptable. The reported barriers, together with reduced motivation and self-efficacy observed in the participants, guided the researchers to develop intervention content focused on improving health literacy in play and development, delivered in the form of an interactive mobile app. Future research will develop and test the efficacy of this intervention in a low resource setting in South Africa.

**Funding:** "This work was supported by the National Institute for Health Research (NIHR) (using the UK's Official Development Assistance (ODA) Funding) and Wellcome [222007/Z/20/Z] under the NIHR-Wellcome Partnership for Global Health Research. The views expressed are those of the authors and not necessarily those of Wellcome, the NIHR or the Department of Health and Social Care." The funders had no role in study design, data collection and analysis, decision to publish, or preparation of the manuscript. AP received funding for this work from the National Institute for Health Research (NIHR) (using the UK's Official Development Assistance (ODA) Funding) and Wellcome [222007/Z/20/Z] under the NIHR-Wellcome Partnership for Global Health Research FB received no specific funding for this work HT received no specific funding for this work SAN received no specific funding for this work Dr Alessandra Prioreschi received a salary from the Wellcome Trust from the same fund.

**Competing interests:** The authors have declared that no competing interests exist.

## 1. Introduction

Early childhood is one of the most important timeframes in life, due to the rapid growth and development of the brain and body during this time [1]. Children are particularly receptive to their environment in the first 1000 days of life, making it an important time for developing healthy behaviours, which encourage positive outcomes later in life [1]. In South Africa, obesity is a major public health concern, with 13% of South African children under 5 years being overweight in 2020 [2], making it important to start encouraging physical activity as soon as infants are born [3]. It is imperative for early movement behaviours (for example, 'tummy time') to be encouraged within the first days after birth, as these activities have been consistently shown to be associated with advanced gross motor and personal-social development over time [4].

In the early months of life, infants require access to unstructured, unrestrictive play at home to encourage development and increased physical activity [5]. This is evident in the poor developmental outcomes seen in children who spend too much time in a supine position as infants, constrained in infant seats, walkers, or in small playpens [5]. Unstructured outdoor play is particularly important as it allows space and opportunities for playful activities, which encourage development of gross motor skills (e.g. kicking and throwing balls, climbing on play equipment, etc.) and higher intensity physical activity required to meet daily movement guidelines [6]. Children need to be provided access to, and opportunities to engage in unstructured play by their caregivers, which highlights the importance of caregiver education and modelling when encouraging more unstructured play with children. By focusing on increasing participation in play from birth, infants and toddlers will naturally be more physically active, helping to reduce the risk of developing obesity and other non-communicable diseases later in life [7].

There are a number of social and environmental factors in the home setting that can limit participation in play, particularly in low-middle income settings such as South Africa. Examples of environmental factors include outside areas which are not considered safe to play (e.g., exposure to violence, heavy traffic, pollution, unsanitary conditions, etc.) as well as inside areas which may be very small or overcrowded with no limitations on screen time and other similar sedentary behaviours [8]. Social conditions that impact play include lack of resources available for toys or books, low maternal health literacy, and poor maternal mental health [8, 9].

As caregivers are their infants' first teachers, they have the important role of providing opportunities and space for children to learn through play [10]. Empowerment and provision of support for caregivers is therefore essential, to allow facilitation of playful learning in day-to-day activities in the home environment and community [10]. This is particularly important in a setting such as South Africa, where caregivers have expressed a lack of motivation and knowledge in how to promote development, particularly through play [8]. Therefore, increasing caregiver health literacy is key to developing a greater understanding of the importance of play and development in the early months [11].

In light of the above, this study aimed to test the acceptability of an intervention developed to improve health literacy around play and development with mothers from the Soweto community in South Africa. The researchers hypothesized that increased health literacy in mothers would improve capability, motivation and opportunity, thereby ultimately resulting in greater participation in infant play. The specific objectives in this paper were:

1. To understand the challenges and barriers that a group of Soweto mothers experience when facilitating play with their young infants.

2. To present a prototype of intervention activities and content, discuss perceptions as to how the activities and content may improve health literacy (thereby encouraging participation in play), and how best to present the intervention.

3. To determine the acceptability of the intervention prototype after being tested by the mothers in Soweto.

## 2. Methods

### 2.1. Study setting and participants

This study was nested in the larger Play Love and You (PLAY) study, which was developed with the overriding theme that all that infants need is Play (opportunities for movement and development), Love (responsive and interactive caregiving) and You (the mother herself) [11]. The PLAY study had three aims. The current study was part of the third aim, namely, "To test the efficacy, feasibility and acceptability of providing content, and increasing opportunities for early learning through play to promote literacy" [12].

In order to develop the intervention, a Community Advisory Group (CAG) was established with the intention of testing the acceptability of the intervention before implementing it in the PLAY Study [12]. Twenty mothers from Soweto with infants aged 0–3 months were recruited in April 2022 to form the CAG. Participants were recruited by approaching women at the taxi rank outside Chris Hani Baragwanath Academic Hospital (CHBAH) to ask whether they would be interested in participating. Mothers were eligible for recruitment if i) they were 18 years of age or older with an infant aged 0–3 months ii) of whom they were the primary caregiver, and iii) were living in Soweto for the duration of the study. Mothers were ineligible if infants were born preterm or if the infant had any disabilities, impairments or disease that would limit movement or impair normal development. 0–3 month old infants were included as this was the age group that would be included in the PLAY study, and as a result of the evidence showing that an increase in movement and play in very young infants is associated with advanced gross motor and personal-social development later in childhood and beyond [4].

**Ethics statement.** Ethical approval was obtained from the University of the Witwatersrand Human Research Ethics Committee (Sub-study M220886 under main study M210846). Written, formal informed consent was obtained from all participants, for participation in the focus group discussions (FGDs) and questionnaires, and for each FGD to be recorded. All participants were reimbursed for travel costs and provided with refreshments.

### 2.2. Data collection

Two rounds of Focus Group Discussions (FDGs) (named "Round 1" and "Round 2", respectively) were conducted for each participant, approximately two weeks apart in April 2022. All FGDs were conducted at CHBAH in Soweto, Johannesburg by staff trained in qualitative research data collection. Although questions were asked in English, participants could speak in their home language and conversations progressed in the vernacular. This was later translated into English if the response was in a different language, and transcribed for analysis. Lastly, Round 3 of data collection, a telephonic questionnaire, was conducted approximately one week after the completion of the FGDs with the same group of participants. All FGDs and telephone calls were audio recorded with the participants' consent and transcribed into English verbatim.

**Round 1.** Round 1 comprised of three FGDs, conducted over 3 days with 8 participants in the first FGD, 6 participants in the second FGD and 6 participants in the third FGD (total

N = 20). All participants started the session by providing socio-demographic information in a questionnaire. The aim of the Round 1 FGDs was to understand the challenges and barriers that the CAG members experienced when facilitating play with their infants. Participants were given a detailed explanation of what the PLAY study was, including the study's aims and objectives, themes, contact points between the different arms of the trial, as well as examples of the types of assessments and interventions that would be included in the study [12]. The theme in the PLAY study that participants were asked to focus on, was around interactive play and promotion of early learning. As a result, discussions were largely around the participants' understanding of play and development, as well as the effect of their home environment on participating in play with their infants. As part of this discussion, participants were shown a number of photos/ pictures depicting different home environments, which invoked a discussion around which environments enabled or discouraged active play. The stock photos were chosen to show a variety of home environments that would be similar to those found in Soweto, including single roomed homes, shacks and hostels, as well as different settings where children may play e.g. on the side of the road, in a yard, etc. Participants also had a general discussion around the delivery of the intervention, e.g. the preferred means of contact and how often participants should be contacted.

**Round 2.** All participants from Round 1 FGDs were invited back for Round 2, approximately two weeks later, however, five of the twenty participants did not attend. Round 2 comprised of two FGDs conducted on the same day, with seven and eight participants in each respective FGD (fifteen in total, n = 15). Round 2 was primarily focused on a discussion around the acceptability of the play and development aspect of the content to be included in the study intervention. Drawing from the themes which emerged in Round 1 as well as previous research conducted in this same population group, we created a 'prototype' or an example of intervention content appropriate for infants aged 0–3 months, which was introduced to the groups in Round 2. This content was in the form of developmentally appropriate activities, which would be delivered to participants as part of the intervention. The content included videos (for example, singing games, clapping hands, rolling games etc), demonstrations of how to make toys from recycling materials (S1 Fig), as well as infographics and other educational material (such as monthly developmental milestones). Participants were asked to give feedback based on their perceptions as to how the activities and content may improve health literacy, as well as how best to present the intervention. Participants were then asked to engage in and complete three specific activities at home. The first activity was based on videos consisting of games to play with baby and trying different tummy time positions. The second activity was for the participants to make a sensory toy, which was demonstrated in person. The last activity was to determine the potential acceptability of the content in the home setting from the perspective of the caregiver. They were encouraged to pick a few that they liked and try those, instead of attempting to complete all of them. The videos were sent to the participants via Whatsapp.

**Round 3.** The participants from Round 2 (n = 15) were then contacted a week later by the study team to complete an acceptability questionnaire, comprising of 24 questions over the phone (see S3 Appendix). The first question determined which prototype activities (sensory square, shaker and five games) had been completed at home. The rest of the questions focused on determining whether the participants found the prototype activities "acceptable" after testing them at home [12]. The acceptability of the activities was determined by asking questions specifically based on the seven component constructs of acceptability by Sekhon et al. [12], which were used to measure acceptability in this study. Examples of questions included, "Were the instructions clear?" (measuring intervention coherence), "Do you think the games are beneficial for you and the baby?" (measuring perceived effectiveness), etc. Each component

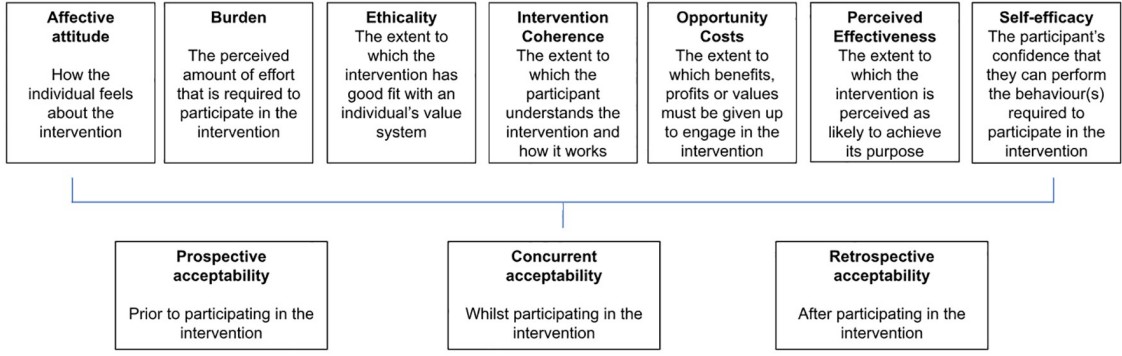

**Fig 1. Definition of acceptability and the seven constructs of acceptability [12].**

construct (namely, affective attitude, burden, ethicality, intervention coherence, opportunity costs, perceived effectiveness, and self-efficacy) is described further in Fig 1.

Fig 2 shows a summary of all three data collection points.

## 2.3. Data analysis

All participants' identities were protected by assigning a unique identifier to each participant, making all qualitative data non-identifiable during the analysis. The quantitative data were de-

**Round 1 (April 2022):**
- Introduction to study
- Introduction to intervention ideas
- Home environment photo discussion

Data collected:
- participants' understanding of play and development, as well as the effect of their home environment on participating in play with their infants.

**Round 2 (May 2022):**
- Participants presented with content prototype,
- Shown how to make sensory toys
- Sent home with prototype activities (in video format)

Data collected:
- Participants' feedback based on their perceptions as to how the activities and content may impact health literacy, as well as how best to present the intervention.

**Round 3: Acceptability Questionnaire (one week after Round 2 FGDs):**
Administered telephonically after activities had been tested

Data collected:
- Acceptability of prototype activities (games, infographics, and making of sensory toys) based on the seven component constructs of acceptability by Sekhon et al. [12]

**Fig 2. Summary of data collection.**

identified at the point of data entry, using the study ID number, so that the researchers could not link the participants to their data. The data used in analysis only had a study ID number, without names or any other identifying information available. The qualitative data were analysed using thematic networks analysis [13]. Meaningful statements and ideas were coded and then grouped into themes. Similar themes were then grouped and placed in thematic networks, which were subsequently summarised and discussed in relation to the original research question. The three transcripts from Round 1, and two transcripts from Round 2 were divided among three researchers who assigned codes to each statement and grouped the codes into themes. The researchers then swapped transcripts to cross-check all the identified themes and codes, which were further checked by a fourth researcher. Thereafter, all researchers met as a group to discuss and refine the themes until no new themes emerged. Means, medians and ranges were calculated on Microsoft Excel and used to describe the demographics of the participants and the quantitative data from the acceptability questionnaire.

## 3. Results

The average (mean) age of the twenty participants was 24 years old and the median was 23.5, ranging from 18 to 43 years old. Eighty percent of the participants described themselves as single, while the other twenty percent reported themselves to be married, engaged or in a relationship. Seventy percent of the participants were unemployed while thirty percent had some form of employment. The majority (sixty percent) of the mothers had completed their Matric (final year of secondary school), while thirty five percent had not completed secondary school. Only one participant had a tertiary degree. Six of the participants were first time mothers. There was a fairly equal distribution of male and female infants, with half being female, forty five percent male, and one infant whose mother preferred not to say. The youngest baby was new-born, and the oldest baby was 5 months old, with the average infant age being 2 months and the median age being 3 months old. The time between enrolling participants and the first FGD was just over a month, which explains why some infants were older than the 3 months inclusion criteria.

### Round 1 & 2

The following eight themes emerged from Round 1 FGDs: 1) Being a mother, 2) Environment, 3) Play and Development, 4) Intervention, 5) Physical Wellbeing, 6) Health Literacy, 7) Breastfeeding, and 8) Mental Health. "Being a mother" included discussions around the challenges and joys that motherhood brings, while the "Environment' theme gave insight into factors in the participants' home and residential areas which impacted their activities, support (or lack thereof), safety and movements. The participants discussed their knowledge and perceptions of the importance of "play and development" in their infants, while the "Intervention" theme largely focused on preferences around the delivery of the intervention. The "Physical Wellbeing" theme explored how the participants manage their infants' and own physical health, and the "Health Literacy" discussion focused on access to information and ability to understand information provided. The 'Breastfeeding' theme was a prominent discussion exploring reasons for breastfeeding (or not), factors impacting continuation of breastfeeding, support available, etc. Lastly, "Mental Health" explored all factors impacting the participants' mental health as mothers. The Health Literacy, Environment, and Play and Development themes were relevant to this study, while the rest are discussed in papers currently in review.

During the second round of FGDs, the three themes which emerged from Round 1 (Health Literacy, Environment and Play and Development) that were pertinent to this study, were discussed again and expanded upon further. Furthermore, one new theme emerged, namely

Acceptability of the intervention prototype content, which was the primary focus of the discussion. The themes were further divided into subthemes and are discussed below.

**Health literacy.** *Clinics/ healthcare workers.* The participants relayed a sense of distrust in the health workers at the clinics. Many reported that they are provided with minimal information around the importance of play or how to encourage normal development, and health workers can be rude or make new mothers feel ashamed for asking certain questions.

*"No you know at the clinic they don't explain anything, they just give you Road to Health [booklet] (a booklet given to parents by clinic staff when children are born which keeps record of the child's growth and other key developmental milestones, as well as providing the parent with information on nutrition, loving & bonding, protection of the child including immunisation schedule, basic health care e.g. treating common illnesses and "extra care" for those living with HIV, disabilities, etc) [14] after giving birth to the baby. You will figure out when you get home when to take the baby in for the three day check-up, and when you have to take them in for six weeks. They don't explain anything. You just have to give yourself time and read the Road to Health book because at the back they have symptoms that explains the things to watch out for in a child. They don't explain. You have to sit and read that book."*

*(Round 1, grp 2).*

*Family/ peers/ other sources.* A few of the participants expressed an interest in receiving additional information on how and when to play with their infants as they did not have access to family and/ friends who could provide them with advice when it came to their infants' development. Participants expressed that even when they did have family members involved, they often did not know who to trust and would prefer objective information from a trusted source. However, they did not tend to trust and/ or understand the objective information given to them (such as the Road to Health Booklet), or did not see its relevance, but did not discuss what/ who they would trust.

*"My mother is there to help me about the baby but I take care of most of the things because my mother loves going out. So most of the time I have to look after the baby so I have to know a lot of things. To clean the belly button and things like that."*

*(Round 1, grp 2).*

*"I would be happy [to receive this intervention], because when you go to the Road to Health [booklet], there are some things which are different, if you go to some pages, they talk about SASSA (An agency that aims to provide comprehensive social security services against vulnerability and poverty) and I don't understand what SASSA has to do with Road to Health.*

*(Round 1, grp 2)*

**Environment.** The discussion around the photos of children playing in different settings evoked various thoughts around safety, space and the effect on the child.

*Physical safety.* The participants were quick to point out that they felt there was a risk of children being kidnapped or subjected to abuse when an area was not enclosed or when there were many men around. There was also mention of children needing to be kept inside or in an enclosed yard to avoid them being involved in motor vehicle accidents.

*"The house is not in a busy part of the township, but you need to keep the gate closed as there is a small child. A car can come and drive over the baby if the baby gets out of the yard"*

*(Round 1, Grp 3)*

*"You know, to be honest, these days, you saw how the first picture was* [picture of children playing on the street] *. . . I won't feel free if my child lives in a place like that. A girl especially, not only a girl, because sometimes boys do get raped as well, you understand, I won't feel happy".*

*(Round 1, grp 2)*

*Effect on development.* Many felt that it was important for children to be around other family members such as grandmothers. This was so that they could also look after the child when the mothers were not able to, without the child experiencing separation anxiety. The participants were also aware of the importance of playing with other children and being around other people for their children's social development. However, there was minimal mention of how the lack of space and outdoor play areas would impact their children's gross or fine motor play and development.

*"For me, I think it is something good* [for a child to play freely], *. . . I did not want him to play outside with other kids. . . so when you close a child up, he can't communicate with other people. I saw right now at school, to get* [him] *to understand a lot of things, it is hard for him, so I think [you need] to give a child the freedom to play. Yeah, it is a good thing.*

*(Round 1, grp 2)*

*In the house: Occupants and space.* The participants were of differing opinions when it came to the condition of a home and how this may impact the child. Many identified that some of the photos showed unsafe conditions for children, such as exposed wires, potential for injury from kettles/ stoves, lack of running water, etc. Others felt that while certain conditions in their own home were unsanitary or unsafe, they were used to living that way (and taking appropriate precautions) as they had always lived in these conditions. It was suggested that any accident that may occur as a result of the home environment was due to the parent/s' negligence.

*"As she says that there is no ideal place where a child can be raised. I feel that the mother can be careless and put water in a dish and not be mindful of the child. If the child falls in the dish and drowns, I feel it would be the mother's negligence, not that the place is not safe for the child".*

*(Round 1, Grp 3).*

Some participants reported that they live in one room houses where the bedroom and kitchen would be combined and felt that the space constraints did have an effect on what activities they were able to do with their children. However, those with bigger houses also expressed difficulties with space due to overcrowding.

*"So did you see the games that you showed us, did you see them, they need open space. I said last time, we live in a four room* [house], *there are a lot of us. There is no time to do these sort of things. When you are sitting like this, you are sitting like this, and that is the end of it.*

*When you move, you don't have space anymore. When you go to the bedroom, they follow you. So the time that you get is when you go outside and sit under a tree and sit on the grass, and play. When it is raining, we are not playing we will just be sitting in the house only".*

*(Round 2, grp 2)*

*Factors hindering play.* The main reasons for respondents not engaging in play with their children included lack of time due to having to go to work, doing chores, and spending time on social media. Some felt that there were no hinderances as they liked to play with their infants, especially as they did not need toys in the first few months.

*"Maybe* [you don't play] *when you are on the phone and on WhatsApp. Sometimes I am on the phone and when the baby is crying I am like wait a bit you, it is WhatsApp. It is when I am looking at statuses".*

*(Round 1, grp 2)*

*"The chores is what I can say when I think about something that is stopping me, because most of the chores are looking at me at home."*

*(Round 1, grp 2)*

**Play and development.** *Activities with infants.* As all the participants had infants below six months of age, they tended to engage in activities which were appropriate for this age range, such as singing and dancing. Some participants also liked to talk to their infants, massage their feet and laugh with them. After seeing a video about ways to engage in "tummy time" with their infants, some participants mentioned that they were not aware that they could put their infant on their tummies in so many different positions.

*"For me, I play with him by singing, and he laughs and enjoys it."*

*(Round 1, grp 2)*

*". . . I play with my child by talking to him. I will talk to him and then sometimes the child shows reactions. He laughs a bit, and then sometimes when I am playing with him, he will look at me but when my other* [child] *is playing with him, he will laugh."*

*(Round 1, grp 2)*

*Maternal beliefs.* Many of the participants were aware of what the term "development" meant, and some were aware that "milestones" were indicators of how their infant was developing. Some felt that play was good for monitoring how their infant was growing and some had insight into the effect of play on motor development.

*"She develops muscles if she plays and you also see the change in her body".*

*(Round 1, Grp 1)*

One of the main beliefs discussed in Round 2 was around tummy time and sleep. Most participants believed that infants slept better when they slept on their stomachs and were not aware of the dangers associated with this.

*"When a baby is sleeping on their tummy, they become better than when they are in other positions and when they're sleeping on their tummy they sleep".*

*(Round 2, grp 1).*

**Acceptability.**   The main purpose of Round 2 FGDs was to determine the acceptability of the prototype and therefore, this theme was only introduced in Round 2.

The acceptability concept was introduced with a general discussion around the intervention activities, specifically how often the content should be delivered and whether it should be delivered through an app or WhatsApp. Access to mobile data (or lack thereof) for delivery purposes was also explored. All of the participants requested a data free option, and some preferred WhatsApp, while others preferred the idea of using an app. The frequency of the content delivery varied, ranging from daily to weekly and bi-weekly (app content).

Regarding the app content, participants did not mind if the videos were in English, as long as there were subtitles accompanying the audio, moving slowly across the screen. As many of the videos available on the internet are usually North American/ Eurocentric, many of the participants suggested using local videos or having actors who were more representative of the Sowetan Community when it came to their race and culture.

*"It should be maybe it could be a black person. You see?*

*Sometimes we as people, we say these things are done by white people."*

*(Round 2, Grp 1)*

After being presented with the prototype activities, the participants further discussed their thoughts and perceptions of each activity. Subthemes were then created to align with the seven acceptability component constructs as described in Sekhon et al. [12] and the discussion points were placed into each subtheme, accordingly. Table 1 expands on subthemes and their supporting quotes.

## Round 3

**Acceptability questionnaire.**   The percentage of participants participating in each respective activity is shown in Fig 3. The results of the acceptability questionnaire and participant's free responses, (separated into each component structure) are reported below.

One participant was not contactable and therefore only fourteen participants completed the telephonic questionnaire. This participant's data were excluded from the final quantitative analysis. The number of participants completing each respective activity is represented in Fig 3. The discrepancy in the number of mothers completing the activities is due to the mothers only being instructed to pick a few activities to complete at home (i.e. they were not required to complete all of them in the week).

Every participant reported that they liked playing the games with their infants, largely due to both the baby and participant's enjoyment, seeing their baby's development and because it encouraged bonding with their infants.

*"Well I like them a lot because I could see that the baby was developing and I was able to see what he likes and what he thought he did not like and he was smiling and showing me that he was enjoying it. That's why I was able to see that he was enjoying the rolling because for me it seemed like the most uncomfortable one but he was enjoying it. And my son is very picky so I*

**Table 1. Subthemes and supporting quotes from Round 2 FGDs.**

| Component constructs | Supporting Quotes |
| --- | --- |
| **Affective attitude (How the individual feels about an intervention)** | |
| Overall, the participants liked the content that was shown to them. They liked that it showed them something they did not know, that it included fathers, that it would help their children grow. | *"What do you think about this paper in front of you, this one of milestones?" "I feel like it's right, it helps". (Round 2, grp 1)* |
| | *"And I also like that they included the father, father figure on the video. . .To show that parenting is not only for the mother, it is for fathers." (Round 2, grp 2)* |
| **Burden (The perceived amount of effort that is required to participate in the intervention)** | |
| The participants felt that the activities (games and crafts) would not take up too much of their time and that they would find time in the day to do the activities with their infants. | *"It won't waste our time. For me it won't waste my time because I also like to teach my baby how we are supposed to play, it won't waste my time." (Round 2, grp 1)* |
| | *"Me too I don't see it as something difficult because sometimes you find that you finished cleaning at about eleven, you have nothing to do, I can do it." (Round 2, grp 1)* |
| **Ethicality (the extent to which the intervention has good fit with an individual's value system)** | |
| This subtheme was not explored as much as the other subthemes, but participants did not show any reason for the content not fitting in with their individual value system. | *"Okay I see that it's right because it's the way we live, the way at home, at the location we live like that, I mean in one room. (Round 2, grp 1)* |
| | *"It is relatable." (Round 2, grp 2)* |
| **Intervention Coherence (The extent to which the participant understands the intervention and how it works)** | |
| The participants understood the prototype activities shown and understood why they were important. | *"It is important to have knowledge about your baby that when they are one month what does your baby do, two months what does your baby do. (Round 2, grp 1)* |
| | *"I am going to do it more often because I only did that tummy time, it was very informative." (Round 2, grp 2)* |
| **Opportunity Costs (The extent to which profits, benefits or values must be given up to engage in the intervention)** | |
| The participants did not seem to mind that there may be opportunity costs as they tended to see the benefit outweighing the "cost". | *"There's nothing that takes time, it's your baby who is important at that time, when they're still young you need to play with them you need to do that, you must. (Round 2, grp 1)* |
| **Perceived effectiveness (The extent to which the intervention is perceived to achieve its purpose)** | |
| As this discussion took place before engaging in the activities it was difficult to determine how effective the participants perceived the prototype to be in the FGDs, but some did acknowledge that the content had been effective in teaching them how to play more with their infants. | *"It helps us, it teaches us new things that we can do with our babies, we did have others, but they also add [to this]." (Round 2, grp 1)* |
| **Self-efficacy (The participant's confidence that they can perform the behaviour(s) required to participate in the intervention)** | |
| Similarly with self-efficacy, it was difficult for the participants to judge their level of self-efficacy in completing the activities in the FGDs, but most seemed confident that they would be able to complete them independently at home. | *"But do you think that that is something that you could make?" "Yes, it sounds easy." (Round 2, grp 2)* |

*was surprised when he enjoyed rolling. The thing about the games basically you are interacting with your baby and bonding in the time we do these games so I don't think there is anything that I did not like about the games."*

–participant with baby aged 5 months.

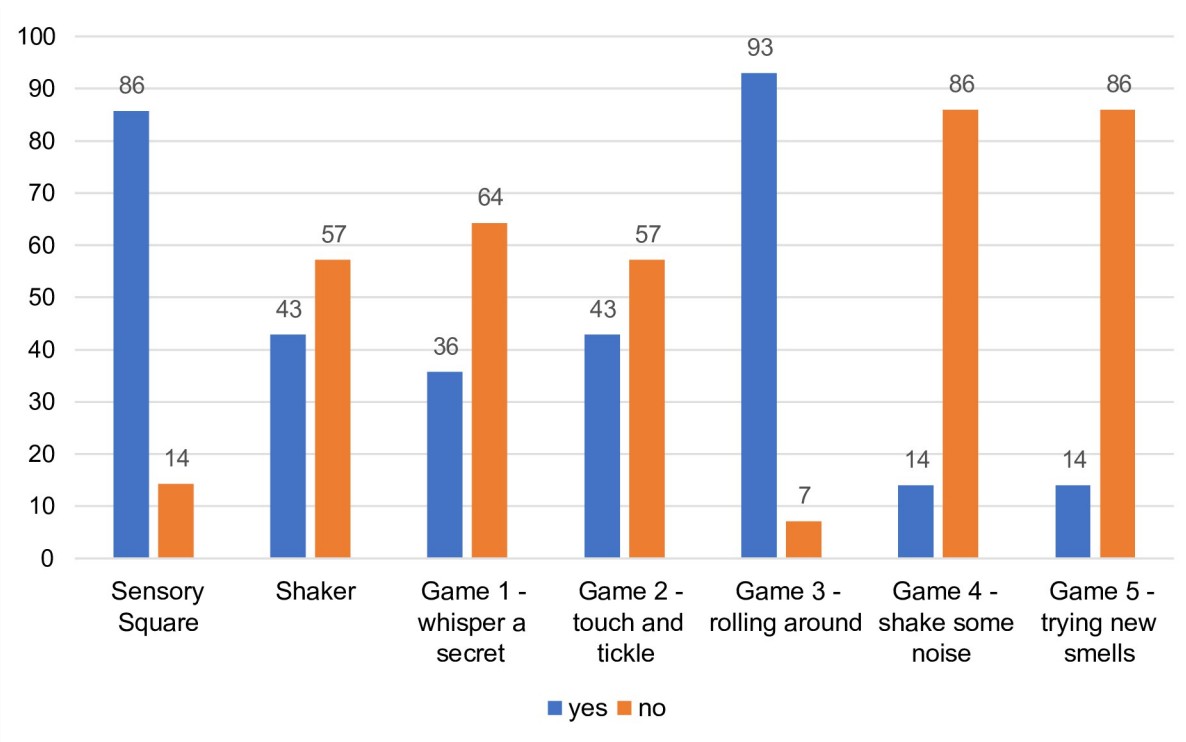

**Fig 3. Percentage (%) of participants completing the sensory toys and video activities at home.**

*"They make her strong, I can see that she is getting used to them. I am also teaching her the tummy to tummy and I have started doing it a lot. She also likes the games but I can also see that some are helping her with development. She can see and follow sounds, that shows me that there is something that she likes and enjoys. There's a lot that I enjoyed about the games".*

–participant with baby aged 2 months.

Only one participant mentioned that they did not like the sensory square because it was too loud for them even though the baby enjoyed it. Otherwise, the rest of the participants stated that they liked the sensory toys because the infants liked the noise, the colours, and holding the square. Other reasons for liking the toys included: using the toys to calm the infant down when they cried, making the infant happy, and encouraging further learning.

*"They make the baby to be more playful when they touch their toys especially the one with the plastic inside* [sensory square] *it makes the baby so happy"*

- participant with baby aged 4 months.

"I loved everything about it, my baby loves it, I can also see that she can hear it and she wants to hold it"

- participant with baby aged 4 months.

*"I loved the sound that it make because now that she is used to it when she cries it calms her down".*

- participant with baby aged 1 month.

Every participant felt that the activities were beneficial. Reasons for finding them beneficial included: enjoying the bonding time, both mom and infant (and family) being happy when engaging in the activities, learning and development for both mom and infant, and safety.

*"[It is beneficial] because they help me to understand my baby and they help her to develop and to learn many things."*

- participant with baby aged 2 months.

*"[It is beneficial] because when I saw the baby smiling and laughing I was also happy so I am happy that they don't only benefit the baby but myself as well"*

- participant with baby aged 1 month.

"[It is beneficial] *because the games are safe"*

- participant with baby aged 2 months.

Every participant found the activities easy to do. The participants had varying reasons for finding them easy. These included the fact that they were easy to understand, that they were bonding with their infants and the infant was enjoying it. The participants agreed that they had the resources needed for making the sensory square or rice shaker. A couple of participants reported that it was easy because they did not need to buy any toys, which is not something they knew beforehand.

*"I found them easy, they were not as hard as I thought they would be because we only think that playing with the baby requires toys."*

–participant with baby aged 1 month.

*"I can sew, so when I was making the toys I was also enjoying it. She enjoyed all of those and they were easy to do because most of the time these are the games that we play with the baby, it's just that we did not know that this is what we must do with our children."*

- participant with baby aged 1 month.

*"They were easy because it was just playing with the baby."*

- participant with baby aged 2 months.

Every participant reported that they understood how to play the games with their infants, however one participant reported that the instructions were not clear. When asked what could be done to make the instructions clearer, the participant responded,

*"Yeah actually I think there should be a person with training to [*show us how to*] play the games sometimes because we forget. So it's easy if we do a practical because other people are slow so it's easy to do a practical for everyone to remember."*

–participant with baby aged 5 months

The majority of the participants (over eighty percent stated that they had time to complete the activities. The remaining two participants were not able to complete the activities due to illness and not knowing that they were required to complete the activities again, respectively. When asked what would help them to make time in future, these two participants stated that they would like to be sent the videos again and reminders through SMS.

All of the participants felt confident that they knew how to do the activities and were doing them correctly. When asked if there was anything that would make them feel even more confident, most of the respondents suggested sending more activities to do, along with instructions on how to do the new activities. Only one participant did not find the toys easy to make.

Of the fourteen participants, just over half (eight) of the participants found it very easy to find the resources to make the toys at home, five participants found it easy to find the resources and one participant found it difficult. While most of the participants had the resources at home, one participant needed to buy a packet of chips to add the packet to the sensory square, and another participant needed to buy a needle, which they could both afford.

When it came to how and when the activities were presented, most (ninety seven percent) of the participants preferred being shown a video instead of it being presented in person, so that they could watch in their own time, slow down and pause the video, and could refer back to it as needed. When asked about how often the participants would like to receive more activities, there were varying responses. Five of the participants wanted to receive content once a week, while three participants wanted to receive the content monthly, every two weeks and every day, respectively.

## 4. Discussion

### Understanding the complexity of the mothers' lived experience

This study aimed to determine the acceptability of a set of activities which would be included in a study intervention for mothers in Soweto. The findings suggest that while many of the participants did enjoy playing with their infants and seeing the impact of play on their child's development, the mothers were also dealing with a range of complex factors, all at the same time, which hindered their participation in play. Some of these factors discussed included lack of time or motivation, environmental constraints as well as limited availability of information from health workers and/ or limited use of the information available, for example, in the Road to Health booklet.

When it came to the mothers' understanding of play and development, many of the mothers were only aware of the gross motor developmental milestones and perceived their infant reaching these milestones to be a good sign of growth. Another study with a similar population group in Soweto, similarly found that mothers were aware of the different developmental milestones and would use them as a tool for tracking development and progress [8]. While the participants understood the value of play in the younger months, it seemed that the participants were not aware of the importance of play in every aspect of development (e.g. not just physical/ motor development), or of the importance of their own role in promoting play with their infants in the early months. Likewise, a Bangladeshi study found that rural mothers had basic knowledge about the benefits of play but lacked insight into the importance of it in every aspect of development, particularly the positive impact on cognitive development [15].

Round 1's discussions highlighted a significant lack of self-efficacy in the mothers, because of their difficult and complex lived experiences. Some of this was due to lack of knowledge and experience but some was linked to difficult circumstances and the environment around them, such as lack of resources and space. This was largely noticeable when many participants showed a sense of excitement and pride in knowing that they could use toys that they had

made themselves for their infant's enjoyment and development, despite their financial constraints. Supporting mothers with education and increased sense of capability, has been shown to have a positive impact on children's development, seen in a study conducted in India, which found that maternal education and mental health were one of the primary indicators for a positive home environment at 6 and 24 months of age [16].

The participants expressed a strong distrust of the health workers at the clinic (who, in South Africa, are predominantly nurse-led). This distrust is largely as a result of the lack of confidentiality and mistreatment by certain health workers, resulting in a distrust in the information provided to mothers in the clinics. Unfortunately, this is something that is widely reported throughout South Africa [17]. Studies conducted in similar settings have reported women feeling insulted by healthcare workers during birth and ante/postnatal care, with health workers being apathetic to their pain or current needs. This resulted in women feeling disempowered to ask for information or not receiving information when asked [18, 19]. As some participants expressed a distrust in information provided by health workers, and others did not trust information given by family or friends, it's clear that there needs to be alternative sources of accurate information that can be relied on. Furthermore, better health literacy in the communities would allow mothers to know where to seek factual, evidence-based information, to know which sources of information to trust, and to know how to apply the information to their own context. These findings guided the researchers to provide intervention content on reliable online play and development resources, as well as how to prepare for clinic visits and meetings with Health Care Professionals.

One of the dominant themes that was repeatedly mentioned, was the impact of the home environment (both inside and outside) on the participants' movement behaviours and activities for both them and their children. Regarding the outside environment, the main concern around physical safety for the child is valid given the surroundings in which the participants live. Soweto is an informal settlement or "township" situated on the southwest of Johannesburg, where the number of child kidnappings, and pedestrian road traffic injuries are high [20, 21]. Similar concerns have been voiced by other mothers in the same Sowetan community. Prioreschi et al. [8] found that mothers expressed concerns around their children being hit by cars, being abducted, kidnapped or raped, as well as other concerns which weren't mentioned in this study but would also be applicable, such as the child hurting themselves or falling into communal toilets. The literature is clear in showing the impact of limited opportunities for movement in home environments (either due to lack of space, overcrowding, etc.) on active play and therefore, motor and cognitive development in infants [22–24]. Most of the participants reported to having limited space indoors and their environment outside being unsafe. This had major implications in the mother's ability to encourage active play, both inside and outside. In view of these circumstances, it was imperative that for any intervention involving active play to be feasible in this community, limited indoor and outdoor space needed to be considered. This resulted in the researchers focusing on increasing health literacy around the value and benefits of play, and specifically how to adapt the home environment, to encourage safe and active infant play.

## Prototype development and engagement

The development and testing of the prototype provided some important insight into how the participants wanted content to be presented. Overall, the participants were happy to try out the activities at home. However, when presenting the activities as part of the prototype, many of the videos or activities did not have a diverse range of ethnicities or socioeconomic groups represented, which the participants indicated would not be their preference. This led to the

intervention only including people, environments and resources that were representative of the Soweto community.

With regards to "tummy time" and sleep related content, many of the participants felt that babies slept better in prone position and were not aware of the associated dangers compared to supine sleeping. This finding is similar to a South African study which found that only 3% of 6-week old babies were sleeping exclusively in a supine position [25] further highlighting the need for greater levels of health literacy within the community, and encouraging the researchers to include tummy time information and associated safety precautions in the intervention activities [13].

## Acceptability of the intervention

Despite the small sample size, the results of the study provided important guidance when creating an acceptable intervention. Firstly, the intervention needed to include simple activities that encouraged age-appropriate play, using very limited resources which were available in the home. Secondly, the activities needed to be culturally appropriate for this specific group, and thirdly, due to crowded living spaces and limited outside play areas, the activities needed to encourage active play within the constraints of small spaces. Lastly, due to the lack of reliable information (and support) available to the mothers, the intervention needed to work alongside the current health system, by providing additional information on play and development that corresponded with information provided by the health care providers. For the activities to be feasible in a real-world setting, activities needed to be simple and achievable, to develop a sense of self-efficacy in the mothers. Content also needed to be accessible given that access to WiFi and data are significant barriers for mothers in accessing information digitally [26].

This study has certain limitations, which the researchers tried to mitigate wherever possible. We did not collect data on how actively mothers truly engaged with the intervention material, and only assessed their perception of a small sample of content. As this study had a small sample size, the findings may not be generalisable, and cannot be extrapolated to higher income settings. However, the sample could still be considered representative of other women in Soweto as certain demographic data, such as parity, education and marital status were similar to demographic data found in other studies completed in the same age group in this area [27, 28]. A common risk when using focus group discussions, is the discussion being led based on the researchers' agenda, or the agenda of specific participants in the group. This risk was minimised by having a trained facilitator who was external to the project and an expert in dealing with group dynamics to allow for equal contribution from each participant. As the author is an outsider to the study population community, with their own ideas and presuppositions, as well as this study being completed as part of a doctoral degree, it is expected that some interpreter bias was present. However, where possible, interpreter bias was minimised by having three researchers code and re-code the data as described in the methods section. Despite these limitations, this study has a number of strengths. Including the participants (end users) from the point of intervention development allowed them to shape and create the health interventions that were appropriate, acceptable and feasible in their own context. This is essential for seeing real success in changing health behaviours in real world settings [29]. Implementing this intervention has the potential to increase mothers' self-efficacy and health literacy around their and the environment's role in their child's play and development. With the right information and tools, each participant has the potential to positively impact their child's development, which if expanded to the larger community, could see large scale positive growth and developmental outcomes throughout the life course. Given the lack of trust in the healthcare system, policy

makers should focus on providing evidence-based knowledge to new mothers through other means such as data free digital products. A focus on providing safe, security patrolled open spaces and playgrounds needs to be prioritized to encourage access to active play for children and adults.

## 5. Conclusion

In conclusion, this study has provided insight into the acceptability of an intervention designed to improve health literacy around developmentally appropriate play during infancy, with a community advisory group of mothers in the Soweto community. The findings presented in this study will guide the development of an intervention in the form of an interactive mobile app and provide further insight and guidance for developing future interventions in low-income settings. Future health and social policies need to account for the complexity of factors and experiences which impact mothers' decisions related to their children's play and development. Further research should focus on the impact and feasibility of using digital technologies to create behaviour change in low-income settings.

## Supporting information

**S1 Appendix. Participant breakdown for each round of data collection.**
(PDF)

**S2 Appendix. Round 1 photos.**
(PDF)

**S3 Appendix. Acceptability questionnaire.**
(PDF)

**S1 Fig. Sensory square.**
(JPEG)

**S1 Data. Participant demographics.**
(XLSX)

**S2 Data. Acceptability questionnaire data.**
(XLSX)

**S1 Checklist. Strobe checklist.**
(DOCX)

## Acknowledgments

The researchers would like to acknowledge the PLAY study research team for their assistance in collecting the questionnaire data and providing administrative support in the Focus Group Discussions. We would also like to acknowledge Lebo Motlhatlhedi for conducting the Focus Group Discussions.

## Author Contributions

**Conceptualization:** Fiona Bennin, Shane A. Norris, Alessandra Prioreschi.

**Data curation:** Fiona Bennin, Helene Theunissen, Alessandra Prioreschi.

**Formal analysis:** Fiona Bennin, Helene Theunissen, Alessandra Prioreschi.

**Funding acquisition:** Shane A. Norris, Alessandra Prioreschi.

**Investigation:** Fiona Bennin, Alessandra Prioreschi.

**Methodology:** Fiona Bennin, Shane A. Norris, Alessandra Prioreschi.

**Project administration:** Fiona Bennin, Helene Theunissen, Alessandra Prioreschi.

**Resources:** Shane A. Norris, Alessandra Prioreschi.

**Software:** Fiona Bennin, Alessandra Prioreschi.

**Supervision:** Shane A. Norris, Alessandra Prioreschi.

**Validation:** Fiona Bennin, Shane A. Norris, Alessandra Prioreschi.

**Visualization:** Fiona Bennin, Shane A. Norris, Alessandra Prioreschi.

**Writing – original draft:** Fiona Bennin, Alessandra Prioreschi.

**Writing – review & editing:** Fiona Bennin, Helene Theunissen, Shane A. Norris, Alessandra Prioreschi.

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
