## [Decision Letter · Decision Letter 0]

25 Jul 2023

PGPH-D-23-01043

Determining the perceived acceptability of an intervention designed to improve health literacy around developmentally appropriate play during infancy, with a community advisory group of mothers, in Soweto, South Africa.

Dear Dr. Bennin,

Thank you for submitting your manuscript to PLOS Global Public Health. After careful consideration, we feel that it has merit but does not fully meet PLOS Global Public Health’s publication criteria as it currently stands. Therefore, we invite you to submit a revised version of the manuscript that addresses the points raised during the review process.

Editor's comments: 

Thank you for considering PLOS Global Public Health as an outlet for your interesting work. I agree with the reviewer's comments. In your response, please make sure to address them carefully. You can ignore the comment about the quantitative analysis. The focus should be on the qualitative results. A sample size of 14 is too small to make any general inference and the descriptive analysis you have conducted is already a stretch. Instead, I suggest using demographic information from these respondents to provide some indication of whether you think the intervention is likely to be acceptable to the wider population (this can be in the discussion section where you talk about the limitations).

I have provided a number of other specific comments in the attached file. Overall, the manuscript can benefit from additional details in several areas and a commensurate trimming of parts that do not speak directly to the research questions. Readers will want to know more about the intervention in particular and the findings on its *acceptability*; the treatment of both of these topics seems insufficient. Please me sure to edit the manuscript thoroughly for structure and clarity, and contact us if you need clarification before you submit a revised version.

We look forward to receiving your revised manuscript.

Kind regards,

Yubraj Acharya, Ph.D.

Academic Editor

Journal Requirements:

2. We have noticed that you have a list of Supporting Information legends in your manuscript. However, there are no corresponding files uploaded to the submission. Please upload them as separate files with the item type 'Supporting Information'. 

3.We have noticed that you have uploaded Supporting Information files, but you have not included a list of legends. Please add a full list of legends for your Supporting Information files after the references list. 

Additional Editor Comments (if provided):

Reviewers' comments:

Reviewer's Responses to Questions

**Comments to the Author**

1. Does this manuscript meet PLOS Global Public Health’s publication criteria? Is the manuscript technically sound, and do the data support the conclusions? The manuscript must describe methodologically and ethically rigorous research with conclusions that are appropriately drawn based on the data presented.

Reviewer #1: Partly

2. Has the statistical analysis been performed appropriately and rigorously?

Reviewer #1: No

3. Have the authors made all data underlying the findings in their manuscript fully available (please refer to the Data Availability Statement at the start of the manuscript PDF file)?

Reviewer #1: Yes

4. Is the manuscript presented in an intelligible fashion and written in standard English?

Reviewer #1: Yes

5. Review Comments to the Author

Reviewer #1: Overall, well done on an very interesting article.

Line 63 – what are the negative effects that may be experienced?

Line 94-95 split the two ideas

Line 97 – there is an updated census please check and use latest statistics

Line 105 – SA is a low-middle-income-setting

Methods

- Were participants in all rounds of the focus group discussions the same as round 1 or were there different participants for each round?

Line 249-250 – this sentence is unclear what is meant by this? Is it referring to the anonymity of participants? Please be more clear this is difficult to understand

More information about the participants should be given what was the exact sample for each round of data collection. Please be specific

Findings

- Please provide insight into the 8 themes identified in the first round

- What is meant by the environment theme each the theme above has an introduction please be consistent throughout the themes and provide insight as to what is meant by the environment theme

What were the themes that were identified in round two? Please help the reader through the findings so that they know which themes will be discussed in each round

Round 2 needs more granular quotes to help the reader understand what the theme is about and what the perspectives of the caregivers were especially the acceptability component. This theme provides limited information about acceptability please provide more insight as to what is acceptable according to the participants.

Round 3

- Provide information as to what the acceptability questionnaire was about

- In the methods, please add a section to highlight what the questionnaire is and what types of questions were asked so that the reader can understand what the questionnaire is about.

Line 507-rephrase one of the participants was not contactable

Overall, 14 participants are quite a small number to draw conclusions as to whether the intervention is acceptable (n=14) participants are not statistically significant. Could the authors please provide insight as to why only 14 participants were included in the questionnaire?

The quantitative component is lacking, and I would suggest highlighting the descriptive statistics and expanding on them in the text.

The discussion should not present any new findings rather just provide a discussion and highlight the similarities and/or difference across the literature. Highlight the strengths and limitations of the study, policy implications and future recommendations

Findings and discussion need more work to help the reader fully understand the findings.

More information about the analysis is required.

6. PLOS authors have the option to publish the peer review history of their article (what does this mean?). If published, this will include your full peer review and any attached files.

**Do you want your identity to be public for this peer review?** For information about this choice, including consent withdrawal, please see our Privacy Policy.

Reviewer #1: No

---

## [Editor Report · Decision Letter 1]

21 Sep 2023

PGPH-D-23-01043R1

Determining the perceived acceptability of an intervention designed to improve health literacy around developmentally appropriate play during infancy, with a community advisory group of mothers, in Soweto, South Africa.

Dear Dr. Bennin,

Thank you for submitting your manuscript to PLOS Global Public Health. After careful consideration, we feel that it has merit but does not fully meet PLOS Global Public Health’s publication criteria as it currently stands. Therefore, we invite you to submit a revised version of the manuscript that addresses the points raised during the review process.

Please address the comments from the reviewer and resubmit. The comments seem straightforward to address. Feel free to ignore the comment about quantitative analysis, as I am aware of the limitations of that analysis. Please read the manuscript thoroughly for flow and clarity.

We look forward to receiving your revised manuscript.

Kind regards,

Yubraj Acharya, Ph.D.

Academic Editor

Journal Requirements:

Additional Editor Comments (if provided):

Please address the comments from the reviewer and resubmit. The comments seem straightforward to address. Feel free to ignore the comment about quantitative analysis, as I am aware of the limitations of that analysis. Please read the manuscript thoroughly for flow and clarity.
---

## [Decision Letter · Decision Letter 2]

7 Mar 2024

PGPH-D-23-01043R2

Determining the perceived acceptability of an intervention designed to improve health literacy around developmentally appropriate play during infancy, with a community advisory group of mothers, in Soweto, South Africa.

Dear Dr. Bennin,

Thank you for submitting your manuscript to PLOS Global Public Health. After careful consideration, we feel that it has merit but does not fully meet PLOS Global Public Health’s publication criteria as it currently stands. Therefore, we invite you to submit a revised version of the manuscript that addresses the points raised during the review process.

We look forward to receiving your revised manuscript.

Kind regards,

Vanessa Carels

Staff Editor

Journal Requirements:

Additional Editor Comments (if provided):

Reviewers' comments:

Reviewer's Responses to Questions

**Comments to the Author**

1. If the authors have adequately addressed your comments raised in a previous round of review and you feel that this manuscript is now acceptable for publication, you may indicate that here to bypass the “Comments to the Author” section, enter your conflict of interest statement in the “Confidential to Editor” section, and submit your "Accept" recommendation.

Reviewer #1: All comments have been addressed

2. Does this manuscript meet PLOS Global Public Health’s publication criteria? Is the manuscript technically sound, and do the data support the conclusions? The manuscript must describe methodologically and ethically rigorous research with conclusions that are appropriately drawn based on the data presented.

Reviewer #1: Partly

3. Has the statistical analysis been performed appropriately and rigorously?

Reviewer #1: Yes

4. Have the authors made all data underlying the findings in their manuscript fully available (please refer to the Data Availability Statement at the start of the manuscript PDF file)?

Reviewer #1: Yes

5. Is the manuscript presented in an intelligible fashion and written in standard English?

Reviewer #1: Yes

6. Review Comments to the Author

Reviewer #1: Overall feedback

Well done on the manuscript. Check the language and sentence structure throughout the manuscript. Overall, the discussion can be shortened and at times there are information that has not be touched on in the findings section. What are the strengths of the study? I am also missing the recommendation for policy, practice and research now that the findings are there what should be done next?

Line 185 – change the N to n

Line 190-193 – please make these two sentences as it is too long and difficult to follow

Line 196 – 200 – Please separate sentence as it becomes a bit difficult for the reader to follow. For example, We asked participants to engage and complete 3 specific activities at home. The first activity was based on the videos which consisted of (). The second activity was for caregivers to make a sensory toy which was demonstrated in person. The last activity was to determine the potential acceptability of the content in the home setting from the perspective of the caregivers.

Line 226 – When interviewing participants, the identities will always be known to us. I think you can delete the sentence and highlight that participants identities were protected by assigning a unique identifier/pseudonym to the participant.

Results

Line 244 – 256 stay consistent with whether you are going to write out the percentage in words or use numerical values to show the percentages.

Line 293 – Make a footnote explaining the road to health as it disrupts the flow of the quote.

Line 335 – motor vehicle accidents should not be capitalized

Line 341 – 345 – This quote can be shortened as most of the information is not relevant the description of the theme.

Line 351- 353 – you can make this into two different sentence to make it easier to follow

Line 359-367 this quote can be shortened and only highlight the most prominent aspects of the effects on development.

Line 411 – you can remove this quote

line 606 – stay consistent with terms you in pervious sections you mention road to health booklet and in this line you use road to health card

line 633 – 635 provide a reference

7. PLOS authors have the option to publish the peer review history of their article (what does this mean?). If published, this will include your full peer review and any attached files.

**Do you want your identity to be public for this peer review?** For information about this choice, including consent withdrawal, please see our Privacy Policy.

Reviewer #1: No

---

## [Editor Report · Decision Letter 3]

7 Aug 2024

Determining the perceived acceptability of an intervention designed to improve health literacy around developmentally appropriate play during infancy, with a community advisory group of mothers, in Soweto, South Africa.

PGPH-D-23-01043R3

Dear Ms Bennin,

We are pleased to inform you that your manuscript 'Determining the perceived acceptability of an intervention designed to improve health literacy around developmentally appropriate play during infancy, with a community advisory group of mothers, in Soweto, South Africa.' has been provisionally accepted for publication in PLOS Global Public Health.

Best regards,

Julia Robinson

Executive Editor